# Learning Curve for Robotic Colorectal Surgery

**DOI:** 10.3390/cancers16193420

**Published:** 2024-10-08

**Authors:** Neng Wei Wong, Nan Zun Teo, James Chi-Yong Ngu

**Affiliations:** Department of Surgery, Changi General Hospital, Singapore 529889, Singapore; teo.nan.zun@singhealth.com.sg (N.Z.T.); ngu.chi.yong@singhealth.com.sg (J.C.-Y.N.)

**Keywords:** robotic, colorectal cancer, learning curve

## Abstract

**Simple Summary:**

Robotic surgery is increasingly being adopted in the field of minimally invasive colorectal surgery and has the ability to overcome the inherent drawbacks of conventional laparoscopic surgery. The adoption of new technology comes with the need to first overcome its learning curve in a safe manner without compromising on patient care. Understanding the learning curve and the factors that affect it can help institutions develop strategies to shorten it and acquire this new skill safely and efficiently. Prior experience in laparoscopic colorectal surgery, robotic surgical simulation, spending time as a bedside first assistant, along with a structured training program with proctorship were found to accelerate the process of skill acquisition. Aspiring robotic surgeons would require as little as 12 cases to attain operative efficiency and 15 cases to reduce complications.

**Abstract:**

With the increasing adoption of robotic surgery in clinical practice, institutions intending to adopt this technology should understand the learning curve in order to develop strategies to help its surgeons and operating theater teams overcome it in a safe manner without compromising on patient care. Various statistical methods exist for the analysis of learning curves, of which a cumulative sum (CUSUM) analysis is more commonly described in the literature. Variables used for analysis can be classified into measures of the surgical process (e.g., operative time and pathological quality) and measures of patient outcome (e.g., postoperative complications). Heterogeneity exists in how performance thresholds are defined during the interpretation of learning curves. Factors that influence the learning curve include prior surgical experience in colorectal surgery, being in a mature robotic surgical unit, case mix and case complexity, robotic surgical simulation, spending time as a bedside first assistant, and being in a structured training program with proctorship.

## 1. Introduction

Globally, colorectal cancer is the fourth most common cancer and the third leading cause of cancer-related mortality according to the World health Organization database [1]. Radical resection has been the mainstay of curative treatment for this disease and can be performed using open (OS), laparoscopic (LS), or robotic (RS) approaches. Over the past three decades, the widespread adoption of laparoscopy in colorectal cancer surgery has improved patient outcomes by reducing pain and wound site complications, hastening the return of gut function, and reducing the length of stay [2]. There are, however, drawbacks to conventional LS, such as impaired dexterity due to a lack of wristed instruments, loss of intuitive movement due to the levering effect of trocars, and operator strain from suboptimal ergonomics. There is also a dependence on a trained assistant for laparoscope control and the provision of retraction [3,4].

RS overcomes many of these shortcomings and has gained popularity amongst colorectal surgeons [5]. The adoption of this new approach is nonetheless associated with a learning curve, represented by the number of procedures required to achieve competency in terms of safety, efficacy, and efficiency [6]. A learning curve analysis will allow institutions to design a robust training curriculum, ensure clinical governance by implementing evidence-based credentialing requirements, and ultimately improve clinical outcomes. From a health economics point of view, it can also be used for cost–benefit calculations when training new surgeons.

In this review, we discuss how the learning curve of robotic colorectal surgery is assessed, the factors that affect it, and how the information derived can be implemented in clinical practice.

## 2. How the Learning Curve in Robotic Colorectal Surgery Is Assessed

Surgical learning curves reflect the acquisition of a skill over time and are measured as a change in a surgical-related variable (surrogate of the surgeon’s performance) over a series of procedures, usually consisting of a graph and/or at least four charted data points.

The classic learning curve in Figure 1 is a sigmoid-shaped curve that consists of three phases. The first phase is one of slow progress before exponentially progressing to the second phase, where execution becomes more fluid with accelerated learning and performance. The final phase is when performance becomes stable as mastery is achieved and additional repetitions no longer contribute to improved performance [7].

An alternative, more commonly described learning curve, as shown in Figure 2, consists of four phases: rapid ascent, slower ascent, plateau or asymptote, and descent [8]. The rapid ascent reflects the initial learner’s growth as they grasp performing a complex procedure before a competency point is reached. Thereafter, additional case experiences improve outcomes but to a smaller extent until a plateau is reached. There may be a temporary deterioration in surgical performance at this point (as shown by the dotted line denoting the secondary learning curve) due to the broadening of selection criteria where the surgeon takes on more challenging cases (patients with more difficult anatomy, higher body mass index (BMI), or previous surgeries). The final phase of descent occurs as the surgeon ages with decreased dexterity and physical capabilities that affect performance outcomes [9].

Learning curves are unavoidable for any surgeon taking up a new skill or technology. By measuring specific outcomes and plotting learning curves, it would be possible to estimate an individual’s level of proficiency.

## 3. Type of Statistical Analysis/Method

Learning curves can be analyzed with a variety of statistical methods, each with its own unique set of advantages and disadvantages, and they are summarized in Table 1.

In its simplest form, a learning curve can be created by plotting the performance outcome (y-axis) against experience (x-axis). A regression analysis can then be performed using various models (e.g., linear regression and least square estimations) to establish causal relationships between the variables, and various curves may be derived (e.g., logarithmic, negative exponential, reciprocal, quadratic, and cubic) [10]. However, there is currently no universal standard for the type of analytic model to use, and defining the level of competency may be challenging. Furthermore, two trainees may have completely different learning curves, making comparisons and generalization difficult [6].

Split group analyses (SGAs) involve chronologically dividing case numbers into fixed groups and comparing outcomes with increasing experience. While ideal for comparing series with large numbers, the splitting of groups is completely arbitrary, and it is difficult to pinpoint when the learning curve is reached—this could occur anywhere between the first and last number of the group.

A moving average analysis is a type of finite impulse response filter used to analyze a set of data points by creating a series of averages of different subsets of the full data set. It is a method for accentuating important trends in the data while playing down the random fluctuations that are invariably present in serial data [10]. Taking the example of operative time, the trend in operation time is obscured by individual variation, and averaging the past values filters this variation and accentuates the trend in the data. Similar to an SGA, the order of the moving average is arbitrarily decided as well (e.g., order of 20/30) [11].

A cumulative sum (CUSUM) analysis is a sequential multidimensional analysis method that detects changes in the individual surgeon’s performance graphically. The curve is constructed as a running total of the consecutive differences between each individual data point and the mean of all data points [12]. The plotted results yield curves that show trends in change and can help identify the point or points at which change occurs [13]. A CUSUM analysis is independent of the sample size and has the ability to allow for a continuous analysis in time and a rapid evaluation of data [14]. It is useful in assessing trends and the possibility of adverse events but not ideal for comparing different surgeons due to the variability in technique and proficiencies.

Risk-adjusted cumulative sum (RA-CUSUM) is an extension of CUSUM where negative variables and/or confounders that might influence a surgeon’s learning curve are accounted for. However, the challenge is the need for large data sets, especially if complication rates are low. Studies on RA-CUSUM estimate each patient’s adverse event risk using a logistic regression model and then update the CUSUM charts using a likelihood-based scoring method [15]. For the RA-CUSUM curves, an upward shift indicates surgical failure, and a downward shift indicates surgical success.

## 4. Variables Used for Analysis

Numerous variables/matrixes that reflect clinical performance can be used to plot learning curves and are summarized in Table 2. Measures of learning related to surgical technique can be broadly categorized into (1) measures of the surgical process and (2) measures of patient outcome [8]. Variables within the surgical process include operative factors like operative time, rate of surgical conversion, blood loss, success of surgery, case complexity, level of proctoring involved, pathological outcomes of resection margins, lymph node yield, and completeness of Complete Mesocolic Excision (CME)/Total Mesorectal Excision (TME). Patient outcomes include intraoperative complications, postoperative complications, the anastomotic leak rate, functional outcomes, oncological outcomes, and mortality.

Surgical process outcomes are typically used for analysis [16] as they are readily accessible. However, they are of lesser clinical significance as they tend to only indirectly relate to patient outcomes—a surgeon who reduces their operative time at the expense of patient outcome would constitute a failure of surgery rather than a success.

Patient outcomes, on the other hand, tend to be dichotomous events like complications or survival for which statistical analyses are more difficult, and they require large data sets if they are rare. Consequently, multidimensional plots which take into account various significant variables are likely to give a more accurate representation of a specific operation’s learning curve [6,8]. Ideally, studies should aim to use hybrid models that incorporate these elements when determining the learning curve to reflect true surgical success or failure. The Global Evaluative Assessment of Robotic Skills (GEARS) checklist is an example of a validated tool containing various domains, including dexterity, efficiency, and robotic control, and it is a more comprehensive assessment of a surgeon’s competency than time improvement alone [9].

## 5. Interpretation of Learning Curve

The aim of evaluating a learning curve is to determine the number of sequential procedures required to reach a certain level of competency. To determine this, studies must define a particular threshold in the surgeon’s performance on the learning curve to indicate competency. However, this threshold is not standardized. Some studies state that it is achieved once a change in phase has occurred (inflection point on the curve—the second vertical dotted line in Figure 2), where others define it as when the plateau has been reached [16,17].

A caveat of assuming competency based on the learning curve is that a plateau in one’s performance may not necessarily equate to reaching a high procedural standard—it could merely imply that the surgeon has stopped improving [6]. Therefore, traditionally, competency has been defined based on a pre-determined threshold set by a panel of experts/senior surgeons [18,19]. Consequently, there remains significant variability in how learning curves are reported, and the generalizability of findings is limited.

While conservative, it is perhaps prudent to assume competency has been achieved when the learning curve has plateaued (indicating stability of performance) and certain expert-derived benchmarks have been met.

## 6. RS Learning Curve in Terms of Operative Time

The most common method of assessing learning curves in the literature is the analysis of time-based metrics such as the total operating time (TOT), surgeon console time (SCT), or procedure-specific time (key operative phases—e.g., splenic flexure mobilization or pelvic dissection phase for TME) [9,16]. The CUSUM methodology is typically used for analysis, and the CUSUM chart is plotted against chronological cases. A typical CUSUM operative time curve is shown in Figure 3.

Using the above figure for illustration, there are three distinctive phases with CUSUM analysis in relation to operative times. Phase 1, which defines the learning phase, is a period of time where there is an increasing trend in operating time (values are above average) as the surgeon becomes familiar with the system (technical aspects like docking, instrument familiarization, troubleshooting, and ergonomics) and starts to train. Upon reaching the first inflection point, the curve plateaus (values are equal to average) to indicate a level of proficiency before possibly having an increasing trend as the surgeon starts to broaden their case selection to take on more challenging cases [20]. The final phase is that of mastery, where the surgeon becomes an expert and the operative time for the same procedure decreases steadily (values become below average). In general, upward and downward sloping plots are indicative of less and more favorable outcomes, respectively.

The surgeon console time is typically used for assessment as it is more reflective of the surgeon’s independent learning compared to the total operative time, which is confounded by many other variables, including the experience of the surgical team.

For this review, we identified a total of 23 full-text articles on learning curves for RS colorectal surgery published from 2010 to 2024. The main characteristics of these studies are shown in Table A1 of Appendix A.

Three studies [20,21,22] performed surgeries using the daVinci Xi robotic system (Intuitive Surgical, Sunnyvale, CA, USA) solely. Another three studies performed surgeries with a combination of the daVinci Xi system as well as the daVinci Si and S systems. The remaining majority of published studies performed surgeries with the daVinci Si robotic system or did not specify the system they used.

All of the studies included time-based metrics as a part of their learning curve assessment, with three studies [15,23,24] additionally incorporating composite outcomes as part of their analysis. The majority of articles evaluated the learning curve for performing RS TME (compared to a variety of operations), with the learning curves ranging from 12 to 74 cases to reach the level of competency. A significant proportion of studies lacked a sufficient sample size to assess the learning curve holistically and provide strong evidence on the mastery phase of the learning curve. The studies that analyzed the mastery phase reported a learning curve of 83–88 cases [20,22,25].

The study that had the largest series performed by a single experienced laparoscopic surgeon to date was by Tang et al. [17], who had a total of 389 consecutive patients over a 4-year period and found the learning curve for RS TME based on a CUSUM analysis of operative time was 34 cases to mount the learning phase and 151 cases for mastery. The surgeon was a pioneer robotic surgeon in his institution.

Guend et al. [12] provided further insights into the differences in learning curves between early adopters compared to later adopters in their analysis of five surgeons in a single institution where the early adopter required 74 cases in comparison to 25–30 cases for later adopters in the initial phase of learning in performing robot-assisted left-sided/rectal resections, advocating the establishment of robotic colorectal cancer programs. Nasseri at al [20] further showed that during each phase of the learning curve, apart from improvement in the surgeon console time, there was also a decrease in the mean length of stay (LOS) for hospitalization.

A systematic review performed in 2016 [26] of seven case series and two non-randomized case comparison series found the mean number of cases of Phase 1 of the learning curve to be 29.7 patients and 39 patients for the surgeon to be classed as an expert.

## 7. RS Learning Curve in Terms of Complication Rate or Patient Outcomes

Other metrics, including the complication rate and patient outcomes, have also been used to measure the learning curve.

Tang et al. [17] factored in postoperative complications of Clavein Dindo ≥3 as an independent variable for their learning curve analysis and found that surgical outcomes improved from the 36th case onwards.

Oshio et al. [15] used RA-CUSUM to factor the complication rate into their assessment and found that surgical failure decreased after 48 cases before stabilizing and decreasing further after 80 cases. This deferred slightly to their CUSUM analysis on operative time, which stabilized after 40 cases.

Shaw et al. [27] referenced prior published studies [28,29] and assessed the learning curve for complications based upon a learning curve of 15 cases and found that the complication rate reduced significantly from 27% to 6.3% after the first 15 cases (*p* = 0.03).

In terms of the impact of training on long-term oncological outcomes, two studies [17,30] found that the learning curve did not have an impact on long-term oncological outcomes for experienced laparoscopic surgeons who were taking on robotic surgery for the first time.

## 8. Learning Curve of RS Compared to LS for Colorectal Operations

With the increasing prevalence of robotic surgical systems, newer generations of surgeons may embark on robot-assisted surgery without first mastering open and laparoscopic techniques. However, the majority of studies that assess robot-assisted learning curves typically involve surgeons with prior experience in open or laparoscopic surgery. Few studies have assessed the impact of a robotic surgical system in training a novice colorectal surgeon.

Quoted learning curves for laparoscopic low anterior resection range from 50–80 cases based on operative times [31,32] to 50–110 cases based on risk adjusted analysis [31]. The learning curve for Transanal TME (TaTME) is quoted to be 30 cases based on the pre-requisite of having advanced laparoscopic skills and prior performance of rectal surgeries independently [33]. If the robotic platform can shorten the learning process, novice colorectal surgeons and patients would not have to bear the consequences of a steep learning curve and its associated complications.

Burghgraef et al. [34] performed a systematic review in 2022 to answer this clinical question but found a significant lack of high-quality evidence and standardized statistical methods regarding the learning curves of the various minimally invasive TME techniques. Of the 7562 patients assessed across the 45 included studies, they found the learning curve to be 50 procedures for laparoscopic TME, 32–75 procedures for robot-assisted TME, and 36–54 procedures for TaTME. The limitations in this study were that the surgeons’ backgrounds were heterogeneous and novice RS was not included in most of the studies.

Melich et al. [35] assessed the laparoscopic and robotic learning curves for low anterior resections simultaneously for a single surgeon that was a novice in minimally invasive colorectal surgery and found that the operative time for robotics, though initially longer compared to laparoscopy, rapidly improved with practice and eventually became faster than laparoscopy after 41 cases. De’Angelis et al. [36] performed a comparative analysis and found that mounting the learning curve for performing a right hemicolectomy in a novice minimally invasive colorectal surgeon was significantly faster for the RS approach at 16 cases versus 25 cases for the LS approach.

Odermatt et al. [37] analyzed the differences in CUSUM graphs for lymph node harvest, length of stay, and major complications for robotic TME, using the surgeon’s matched laparoscopic reference group as a baseline, and found that the learning process for robotic TME was shorter to reach a similar performance level obtained in conventional laparoscopy.

Kim et al. [38] evaluated whether prior laparoscopic experience was mandatory in overcoming the learning curve for RS and found that a single-step transition from open to robotic rectal surgery can be achieved without extensive prior laparoscopic experience. Their study quoted the requirement of 17 cases with a moving average curve to attain a change in the curve.

That being said, the majority of existing studies with quoted shorter learning curves of 13–20 cases [20,27,28,37] for RS are typically conducted on surgeons with prior advanced laparoscopic skills, suggesting that having prior laparoscopic experience resulted in a shorter RS learning curve.

We conclude from the above that a novice surgeon would mount the learning curve in performing a particular colorectal procedure (e.g., TME) faster for the RS approach compared to the LS approach. However, experienced laparoscopic surgeons will mount their RS learning curve faster than their novice counterparts.

## 9. RS Learning Curve for Colon (CME)

The bulk of the existing literature has been concentrated on mounting the learning curve for robot-assisted rectal (TME) surgery. As robotic platforms become more commonplace, we foresee that the future generation of colorectal surgeons who take up robot-assisted surgery may first mount their robotic learning curve on an easier operation like colonic resection before moving on to perform more complex procedures like low rectal surgery with TME. It is therefore imperative to understand the learning curve of robotic colectomies as well.

We found two studies that evaluated the learning curves of performing RS right hemicolectomies [36,39]. Huang et al. [39] found that the learning curve for robotic right hemicolectomy based on the operative time via a CUSUM analysis was 27 cases based on a retrospective analysis of 76 cases. De’Angelis et al. [36] found the learning curve to be 16 cases based on the operative time via a CUSUM analysis of a novice fellow that performed 80 consecutive right hemicolectomies—30 robotically and 50 laparoscopically (the order of cases was not specified).

Other studies that evaluated the robotic learning curve of colorectal surgeries [12,20,27] included cohorts that comprised a variety of colorectal procedures (colonic and rectal), making procedure-specific analysis difficult despite the fact that they reflect a learning curve that is closer to reality.

## 10. Factors That Affect the Learning Curve

Learning curves are influenced by numerous confounders. Understanding how some of these factors can affect the learning curve is crucial in practical application and helps us appreciate why learning curves may differ between institutions and individuals.

### 10.1. Prior Surgical Experience

A surgeon’s prior surgical experience has been shown to be a significant factor for those taking up a new surgical technique/technology. Soomro et al. [16], in their systematic analysis, found that those with greater experience required fewer procedures to overcome the robot-assisted learning curve.

In particular, as mentioned earlier, surgeons who had prior laparoscopic experience were shown to have shorter learning curves for RS compared to novice surgeons. This suggests that there are transferable skills, and those who had originally embarked on a laparoscopic journey can still adopt robotic surgery into their practice at a shorter learning curve.

Shu et al. [40] found that surgeons with extensive laparoscopic experience who underwent a structured robotic training curriculum had a shorter operation time (200.9 vs. 254.2 min; *p* < 0.001) and fewer intraoperative complications (2.7% vs. 21.4%; *p* = 0.015) and postoperative complications (6.7% vs. 22.7%; *p* = 0.025) compared to their counterparts with no prior laparoscopic experience for the first 15 robotic surgery cases. They speculated that this was likely attributed to better recognition of anatomical landmarks from prior experience and the familiarity of operating within a smaller space.

### 10.2. Institutional Factor

Guend et al. [12] reported that surgeons who were in institutions that had an established training program were able to overcome their learning curves in robot-assisted colorectal resections with a smaller case volume (25–30 procedures) compared to an earlier surgeon (74 procedures) who had joined prior to the establishment of the formal program [12]. This concept of institutional learning is not new and reflects that as a surgeon grows, the team around the surgeon also familiarize themselves and benefit from the experience of performing surgeries with the primary surgeon. This familiarity reduces the total operative time by reducing the robotic docking times and the speed in troubleshooting malfunctions in the systems and equipment. Therefore, the learning curve does not only apply just to the surgeon but also includes the experience of the entire operating team as well.

Surgical protocols are also developed over time as the first adopter mounts their learning curve. Adhering to clear surgical protocols guarantees consistency in surgical techniques and minimizes variability, ultimately leading to comparable surgical and oncological outcomes with subsequent adopters. Efforts should consequently be concentrated on training the entire surgical team rather than just individuals for a new service/program to succeed [41]. Institutions who are planning to start on a robotic surgical program should look towards sending a team of individuals together with the surgeon when they learn from more established expert centers.

### 10.3. Variety of Colorectal Operations/Case Mix

Not all types of colorectal surgeries are equal in terms of difficulty and their learning curves [42]. In a practical setting, most surgeons that adopt a new technology usually train on consecutive cases that are heterogenous in nature (e.g., CME, TME, D3 dissection, and intracorporeal anastomosis (ICA)), each encompassing its own set of challenges and nuances. Despite these variations, there are certain transferable skillsets that are attained with each consecutive procedure and can confound the learning curves if they were just assessed based upon one procedure (e.g., TME dissection), i.e., studies conducting a learning curve assessment of one particular operation may not truly be on consecutive surgical cases for that surgeon.

### 10.4. Case Complexity

In addition, each case is unique with its own set of challenges. Certain common surgical or patient factors that typically bring on an added challenges for colorectal surgeons include prior surgeries, the use of neoadjuvant chemo-radiotherapy, individuals with a high BMI, a narrow pelvis/male pelvis, and bulky or locally advanced tumors.

Consequently, if a surgeon takes on a more difficult case (e.g., minimally invasiveHartmann reversal) in a patient with a challenging body habitus, the learning curve will be further skewed.

It has been observed that following the stabilization of performance after mounting the initial learning curve, there is typically a decline in performance [7]. This decline has been attributed to the surgeon taking on more challenging anatomy and complex procedures after mastering the basics of a procedure.

### 10.5. Technology—daVinci Si vs. Xi vs. V

Recent studies published on RS learning curves have typically assessed the learning curve of surgeons using the daVinci Si system [15,30]. As mentioned earlier, of the 23 articles, only 3 used the daVinci Xi solely, and another 3 used a combination of Xi and Si/S.

The da Vinci Xi system, introduced in 2014, is a boom-mounted platform that is capable of multi-quadrant surgery and is less onerous to set up when compared to the Si/S systems. In one study, the daVinci Xi system was found to be an independent factor associated with a reduced surgeon console time compared to the Si system in a multivariate analysis [43]. The daVinci Xi system was further shown to allow for a totally robotic approach to be used instead of a hybrid approach that many have adopted when operating with the daVinci Si system [44]. The limitations for studies which compared the outcomes of the Xi and Si systems, however, relate to the influence of chronology bias and the proficiency gain effect [22,44].

The latest iteration from Intuitive, the daVinci V system, with its ability to provide haptic feedback, is postulated to further reduce this learning curve which, in the past, had depended only on visual cues—a skillset that has its own learning curve. Shu et al. [40] found that the majority of their intraoperative complications in the initial part of their learning curve were intestinal and vessel injuries which they attributed to a lack of tactile feedback on the robotic platform and required the experience of using visual cues to moderate the amount of force applied to tissues during surgery.

### 10.6. Surgical Simulation

The majority of the current RS curriculum and present international accreditation require some form of prior experience in a surgical simulation laboratory. A surgical simulation allows for the acquisition of specific robotic surgical skills without the need to practice on live patients. Robotic surgery requires multi-tasking and coordination of all four of the surgeon’s limbs in comparison to conventional laparoscopy which usually only requires the use of one lower limb for foot pedal activation on top of both upper limbs. There is a need to practice and rehearse on a simulator to ensure that changes in instrument activation, adjustments in optics, and the activation of surgical energy are seamless prior to practicing on live patients.

Culligan et al. investigated the impact of simulator training on performing supracervical hysterectomy on live patients and found that novice robotic surgeons who trained on a virtual simulator for an average of 20 h outperformed (with regard to time, blood loss, and blinded video assessment) a control group with no prior simulator exposure [45]. The learning curve may be shortened by prior practice on the simulator as well as by sequential modular learning of different steps in an operation.

### 10.7. Time Spent as First Assistant

Most surgical training curriculums are sequential and require trainees to progress from first being a bedside assistant before becoming the primary surgeon. Robotic surgery training is no different to this, especially for someone who is not a pioneer adopter of robotic surgery in their institution. The benefit of first being a bedside assistant is having experience in troubleshooting management, collision (external/internal) avoidance, and seeing how the robotic patient cart interacts with the operative environment. This experience and awareness can further help the console surgeon guide their bedside assistant in future to avoid collisions with robotic arms [46].

Cimen et al. [47] found that prior bedside assistant experience for robotic prostate surgery was associated with reduced surgeon console time and median length of stay despite having more challenging cases when the assistant progressed to be the console surgeon. Favre et al. [48] further substantiated this and showed that prior bedside assistance allowed the surgeon to bypass the time-based learning curve for RS hysterectomy.

### 10.8. Structured Training and Proctorship

Having a structured training curricula has been shown to help surgeons make a smooth transition from laparoscopic to robotic surgery [40]. Well-constructed robotic training programs can help the surgical team mount their learning curves in a short time [49]. However, there is no global standardization for robotic training [40,50].

Current evidence is unanimous regarding the importance of having an experienced proctor [24,51]. Proficiency-based curriculum as well as mentorship were further found to result in decreased operative time, conversion rates, and blood loss for surgeons mounting their learning curves for robotic pancreaticoduodenectomies [52].

As we can see from the above, trying to account for the many variables that contribute to one’s learning curve is challenging. Many studies fail to describe the baseline characteristics of the surgeons, patients, or methods of assessment in sufficient detail to make valid comparisons between studies and enable reproducibility.

## 11. How Information from Learning Curves Can Be/Have Been Used

The implementation of new techniques/technologies is always challenging in healthcare, and mounting the learning curve without compromising on patient care is paramount in the early stages of any program in setting it up for success.

Studies have shown that with a structured training protocol, the quality of robotic surgery between a novice surgeon and an expert surgeon can be comparable [53]. The current training curricula for robotic surgery is not standardized and differs between institutions. The sequential stepwise progression of a robotic surgeon’s journey involves a combination of foundational learning on the robotic system, robotic surgical simulations (dry lab), training on cadavers or animal models (wet lab), case observation, supervised surgery under proctorship in the initial phase (modular training in steps; proctored training with/without dual console), and independent practice before achieving mastery and being a proctor. Soliman et al. [54] devised a four-phase training protocol for residents in colorectal surgical fellowship as a framework on logical progression in training. The European Society of Coloproctology recently published its guidelines on training in robotic colorectal surgery [19] this year as well, providing 11 recommendations based largely on expert opinion in the hopes of providing some guidance on training in robotic colorectal surgery.

Knowledge about the learning curve can help ensure that for cases performed in the initial phase, a trained proctor should be present for mentorship and, ultimately, patient safety [40]. Units that intend to start a robotic program can search for a reputable proctor and send their operating team to participate in cross-hospital exchanges to reduce the learning curve during the initial phase, so that when they start, the team would be at a level closer to competency. The learning curve can further aid institutions in identifying surgeons in their mastery phases who may be suitable proctors to guide future generations through knowledge sharing. Zahid et al. [55] outlined in their review the attributes of a high-quality contemporary proctorship program.

The above elements guide the governance of credentialing standards in institutions where other objective domains are assessed in concert with case numbers (which is a crude measure of competency) before providing robotic accreditation to their surgeons. Training programs could also tract the progress of their residents/trainees based on prior learning curve data.

Understanding the learning curve can further help explain the limitations in the current evidence. The pivotal ROLARR randomized controlled trial, which found no differences in the outcomes between RS and LS, only required surgeons to have 10 prior cases as a quality standard and was criticized in view of this as robotic surgeons in this study may have still been in their initial phases of the learning curve compared to the laparoscopic arm, which included surgeons who were experts in conventional laparoscopic surgery (the standard of care at that time), and it therefore underestimated the full potential benefits of robotic assistance [56].

Robotic surgery remains costly and inaccessible to many. Determining the most cost-effective approach that maximizes value and quality at the lowest costs is paramount. Institutions planning to adopt robotic surgery need to factor in the economic impact of the learning curve, where the costs of training and suboptimal outcomes contribute to the financial burden of the health system, and assess that there is sufficient clinical load to justify the cost of the robot.

## 12. Current Limitations in Learning Curve Assessment

There is significant heterogeneity in how the performance thresholds of learning curves are defined as some define performance when a plateau is reached, whereas others use the point of first inflection. There is also no widely accepted and validated model to accurately quantify this point, and some studies merely use visual fit alone [16].

A CUSUM analysis, which is typically used in the assessment of surgical learning curves, is not a perfect statistical method as well and has its limitations where the inflection point/peak of the CUSUM curve is influenced by the total number of procedures evaluated due to its mathematical properties (the higher the total number of procedures, the higher the number of cases required to reach “peak performance”) [57].

There is a need for studies with appropriate evaluation methods, standardized terminology, and necessary context for robust comparisons to be made. Only then can better estimates of learning curves be provided to enhance surgical training programs and improve patient outcomes [16].

In addition, most studies performed are retrospective case series from a single institution and focus on a single surgeon’s learning curve, which may limit generalizability.

## 13. Conclusions

The understanding of a learning curve analysis and the factors that influence it has an impact on training curriculum, clinical governance, and, ultimately, patient safety and outcomes. An ideal statistical method for a surgical learning curve analysis and its interpretation is still lacking.

Aspiring robotic surgeons would require as little as 12 cases to attain operative efficiency and 15 cases to reduce complications. Prior experience in laparoscopic colorectal surgery along with a structured training program with proctorship would accelerate the process and ensure patient safety during the learning curve. A successful robotic colorectal program is dependent upon many factors that extend beyond the surgeon’s training console.

## Figures and Tables

**Figure 1 cancers-16-03420-f001:**
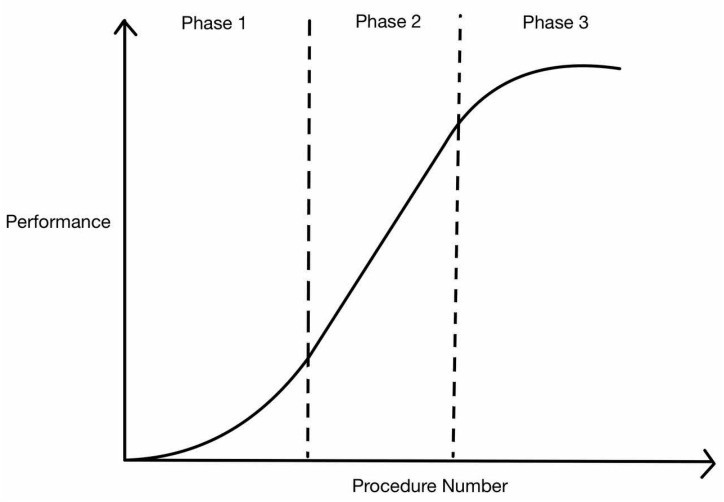
Classic learning curve.

**Figure 2 cancers-16-03420-f002:**
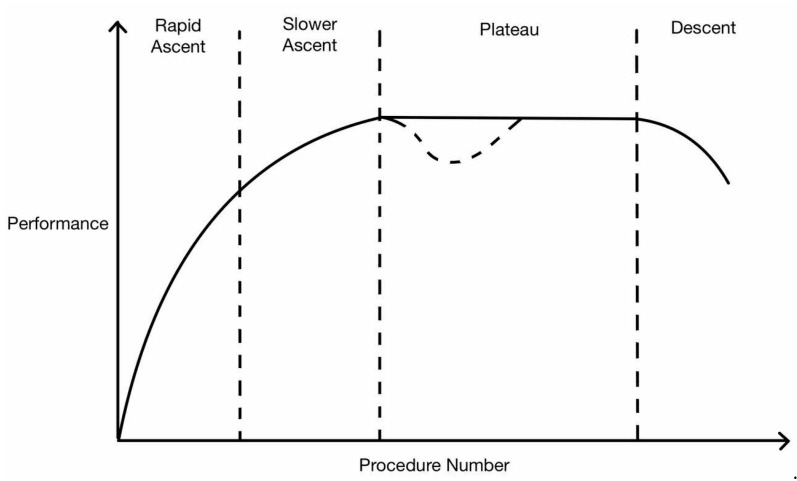
Alternative learning curve.

**Figure 3 cancers-16-03420-f003:**
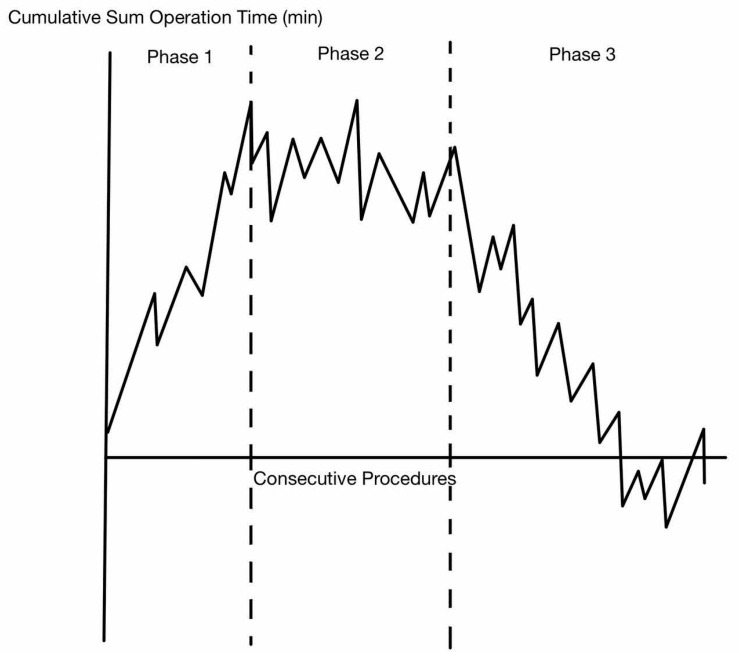
An example of a CUSUM chart.

**Table 1 cancers-16-03420-t001:** Types of statistical analyses used to analyze learning curves.

Type of Statistical Analysis	Advantages	Disadvantages
Regression analysis	Easy to perform	Oversimplified
Split group analysis	Able to compare large case numbers	Split between groups arbitrary with no rationale for cut-off point Difficult to pinpoint exact number required to overcome learning curve
Moving average analysis	Simple and easy analysis of consecutive cases Decreases random fluctuations that occur with serial data	Can only be used for operating time analysis Order of moving average is arbitrarily decided
Cumulative sum (CUSUM) analysis	Detects change in individual surgeon performance	Does not take into account heterogeneity of cases and different case complexities Influenced by total case number assessed
Risk-adjusted cumulative sum (RA-CUSUM) analysis	Ability to correct for case mix that may influence the risk of an event	Difficult to perform Requires large data sets, especially if negative variables/confounders are rare events

**Table 2 cancers-16-03420-t002:** Variables used to analyze learning curves.

Indicator of Surgical Performance	Examples of Variables
Time	Total operative time Console time Time taken for each surgical phase
Intraoperative morbidity	Injury to bladder/urethra/ureter/vagina/intestine Bleeding requiring transfusion Conversion
Postoperative morbidity	Clavien Dindo grade 2 or higher Reoperation
Pathological outcome	Resection margin positivity Incomplete TME Lymph node yield
Functional outcome	International prostate symptom score International index of erectile function Quality of life
Composite	Combination of above variables

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
