# Peer review of "Learning Curve for Robotic Colorectal Surgery"

_cancers, 2024, doi:10.3390/cancers16193420_

Round 1

Reviewer 1 Report

Comments and Suggestions for Authors

Learning Curve for Robotic Colorectal Cancer Surgery

This comprehensive review article has been well written and divided under appropriate subheadings.

Other comments:

1.      The review examines the learning curve for robotic colorectal surgery and therefore the title alluding to cancer is imprecise.

2.      The abstract does not summarise the review article well.

3.      The introduction mentions no compromise of laparoscopic surgery on oncological outcomes – which is not completely correct for rectal cancer.

4.      Under the variables section, multidimensional analysis may be more correct than analysis of multivariate plots.

5.      The authors should incorporate the findings of Lin et al regarding the CUSUM curve peak.  Lin PL, Zheng F, Shin M, Liu X, Oh D, D'Attilio D. CUSUM learning curves: what they can and can't tell us. Surg Endosc. 2023 Oct;37(10):7991-7999. doi: 10.1007/s00464-023-10252-1. Epub 2023 Jul 17. PMID: 37460815; PMCID: PMC10520215.

6.      With regards to factors that affect the learning curve: prior surgical experience should include a section on open colorectal surgery experience, institutional factor can include mention of standardised surgical protocol and dedicated team, case mix section can consider emphasising that the studies which look at one operation (e.g. TME) are not truly consecutive cases and therefore skew the learning curve.

7.      Case mix and case complexity discussion should be separated.

8.      There is no mention of hybrid surgery which can significantly impact the learning curve.  Some studies mention laparoscopic mobilisation of the colon as part of robotic surgery.  It can be predicted by the ratio of console time to the total surgery time, with a lower ratio indicating a larger component of laparoscopic or open surgery.

9.      Cognitive training is another factor which may influence the learning curve.

Reviewer 2 Report

Comments and Suggestions for Authors

Thank you for the opportunity to review the manuscript.

It presents an important topic of modern medicine and aims to provide an overview regarding the fundamentals of learning 18 curve analysis, how it is interpreted and its current application in colorectal cancer surgery with  focus on robotic-assisted surgery in this field.

Please find below my comments:

1. Title is clear and concise

2. The abstract is narrow and does not properly represent the study. Please elaborate

3.Even though, it is a narrative review (not systematic) I strongly suggest including the methods section in the manuscript. In the methods please describe the literature search strategy, inclusion, and exclusion criteria,

4. A flow-chart explaining the search strategy would be very helpful.

5. The manuscript's visual materials are interesting and informative.

6. Please add possible research and clinical implication of this study results.

7. This is very interesting article, however, the conclusion is not properly presented, it is narrow and should be expanded.
